# Sophorolipids—Bio-Based Antimicrobial Formulating Agents for Applications in Food and Health

**DOI:** 10.3390/molecules27175556

**Published:** 2022-08-29

**Authors:** Wei Yan Cho, Jeck Fei Ng, Wei Hsum Yap, Bey Hing Goh

**Affiliations:** 1School of Pharmacy, Faculty of Health and Medical Sciences, Taylor’s University, No. 1 Jalan Taylors, Subang Jaya 47500, Malaysia; 2Centre of Drug Discovery and Molecular Pharmacology (CDDMP), Faculty of Health and Medical Sciences, Taylor’s University, Subang Jaya 47500, Malaysia; 3School of Biosciences, Faculty of Health and Medical Sciences, Taylor’s University, No. 1, Jalan Taylors, Subang Jaya 47500, Malaysia; 4Biofunctional Molecule Exploratory Research Group, School of Pharmacy, Monash University Malaysia, Bandar Sunway 47500, Malaysia; 5College of Pharmaceutical Sciences, Zhejiang University, Hangzhou 310058, China

**Keywords:** sophorolipids, biosurfactants, glycolipid biosurfactants, applications, uses

## Abstract

Sophorolipids are well-known glycolipid biosurfactants, produced mainly by non-pathogenic yeast species such as *Candida bombicola* with high yield. Its unique environmental compatibility and high biodegradable properties have made them a focus in the present review for their promising applications in diverse areas. This study aims to examine current research trends of sophorolipids and evaluate their applications in food and health. A literature search was conducted using different research databases including PubMed, ScienceDirect, EBSCOhost, and Wiley Online Library to identify studies on the fundamental mechanisms of sophorolipids and their applications in food and health. Studies have shown that various structural forms of sophorolipids exhibit different biological and physicochemical properties. Sophorolipids represent one of the most attractive biosurfactants in the industry due to their antimicrobial action against both Gram-positive and Gram-negative microorganisms for applications in food and health sectors. In this review, we have provided an overview on the fundamental properties of sophorolipids and detailed analysis of their applications in diverse areas such as food, agriculture, pharmaceutical, cosmetic, anticancer, and antimicrobial activities.

## 1. Introduction

Biosurfactants are surface-active compounds that are produced by microorganisms such as bacteria and yeasts. The prominence of these biomolecules is reflected through their unique properties including biodegradation capacity, low toxicity, improved environmental compatibility, and enhanced specific activities under extreme pH, salinity, and temperature [1]. Therefore, biosurfactants produced by microbes have been shown to be involved in wide ranges of applications such as food, pharmaceutical, agriculture and cosmetics due to their diversity and remarkable functional properties [1]. Additionally, due to the amphiphilic nature of biosurfactants, it reduces the interfacial and surface tension between phases and results in micelle formation. With these features, biosurfactants can be used as emulsifiers, foaming agents, wetting agents, as well as detergents, thus making them potential agents in industrial sectors such as bioremediation and microbial enhanced oil recovery [2]. Based on Thakur et al. [3], biosurfactants have been divided into different categories such as polymers, lipopeptides, polypeptides, glycolipids, and many others [3]. Among all, glycolipid biosurfactants are currently one of the most broadly studied biomolecules with outstanding properties in applications for food and health (Figure 1).

Sophorolipids are glycolipid biosurfactants composed of a hydrophilic sophorose unit and a hydrophobic fatty acid tail. In general, crude sophorolipid consists of both lactonic and acidic forms of sophorolipid that are structurally different in terms of their length of the fatty acid chains, degree of unsaturation, and types of acetylation depending on the substrates used in the process of production [4]. The structural variation causes a marked discrepancy in biological and physicochemical properties [5]. Due to their low toxicity and high biodegradability properties, sophorolipids are currently explored to be applied as biosurfactant in various aspects such as reducing surface and interfacial tension, increasing the dissolution rate of hydrocarbons, facilitating solubilization and absorption of compounds [6]. Sophorolipids are one of the most attractive and promising biosurfactants found in nature and have been shown to have various biological activities such as antimicrobial, anticancer, antiviral, as well as immunomodulatory activities [7]. The production of sophorolipids from *Starmerella bombicola* yeast species has become an interest in the application of cosmetics, pharmaceuticals, food, and health industries due to their source of many active substances, for example, carbon sources such as glucose and fatty acids [7]. Additionally, sophorolipids are natural compounds, which can rarely cause undesirable side effects, making them a great attention focus in current research.

Sophorolipids that are synthesized by non-pathogenic yeast species have gained a lot of attention and interest in recent years, mainly due to their unique properties of an environmental-friendly nature. The demand for sophorolipids is increasing for various end-use sectors such as food and health industries, hence contributing to the rising market for sophorolipids [8]. As such, there is an increasing trend in the number of publications and research activities of sophorolipids related to its wide range of applications over the years. Moreover, the growth of the publications on sophorolipids has resulted in an increased number of review articles. Considering human health and the environment, the interest in the research on sophorolipids has increased and has developed a strong interest in the unique properties of sophorolipids and explores their vast potential applications. 

According to Akubude et al. [9], the introduction of sophorolipids in the food industry has shown beneficial effects in food processing and food products because of their positive impacts on human wellness and the natural environment [9]. They appear to be promising compounds used in substituting the usage of chemical surfactants which will cause detrimental effects on human beings and the environment [9]. Further studies show the utilization of sophorolipids in the agriculture field to enhance crop yield and to protect against plant diseases [10]. From this review, sophorolipids serve as potential biopesticides as they exhibited strong antimicrobial activity against phytopathogens such as bacteria and fungi, making them useful in the agriculture field in recent years [10].

Generally, sophorolipids play significant roles as biosurfactants in food and health industries, demonstrating a great variety of potential applications with their unique functional and structural characteristics. Sophorolipids have currently been applied and utilized in diverse areas. Their ability to biodegrade and the properties of being environmentally friendly make them a very promising and advanced scientific research topic, especially in the food, pharmaceutical, agriculture, and cosmetics industries. Thus, in this review, we explore the current state of research areas in sophorolipids as well as potential future research.

## 2. Overview of Sophorolipids

### 2.1. Origin and Structure of Sophorolipids

Sophorolipids were originally discovered by Gorin et al. in 1961 [11]. Sophorolipids are an extracellular glycolipid mixture which is produced by *Torulopsis magnoliae* [11]. In 1968, Tulloch et al. also reported that extracellular glycolipids were synthesized by the yeast *Candida bogoriensis*, later as renamed *Rhodotorula bogoriensis* [12]. Sophorolipids produced by *Rhodotorula bogoriensis* and *Candida apicola* have the same structure, but the ones synthesized by *Rhodotorula bogoriensis* have a sophorose unit attached to 13-hydroxydocosanoic acid [13]. Thereafter, the third yeast strain of sophorolipid was discovered by Spencer et al. from the species *Candida bombicola* and the properties and production of glycolipids of this yeast species were similar to those of *Candida apicola* [14]. Furthermore, yeast species *Starmerella bombicola* was found to synthesize sophorolipids and it was presented as the teleomorph of *Candida apicola* as they could generate ascospores through sexual reproduction based on the high 18S recombinant DNA (rDNA) identity between two yeast strains [15]. Additionally, Chen et al. [16] discovered that sophorolipids can also be produced by yeast strain *Wickerhamiella domericqiae* and the major glycolipids observed were nearly similar to those produced by *Candida apicola* and *Candida bombicola* [16]. Sophorolipids have been produced by various non-pathogenic yeasts, especially *Candida* species, including *Candida bombicola*, *Candida batistae*, *Candida apicola*, *Rhodotorula bogoriensis*, and *Wickeramiella domerquiae* [17]. The formation of sophorolipids by non-pathogenic microorganisms makes them advantageous in the application of food and health-related industries [18].

*Candida bombicola* is the most intensively studied yeast strain that produces sophorolipids and it is widely known due to its high efficiency for sophorolipid production [19,20]. Sophorolipids are produced mainly by *Candida bombicola* ATCC 22214, which originated from genus *Torulopsis.* Yeast strain *C. bombicola* was originally isolated from bumblebee honey and it was termed as *Torulopsis bombicola* due to its close correlation with bumblebees [21]. Later in 2012, the yeast species was renamed *Starmerella bombicola*, which is the teleomorph of *Candida bombicola* [15]. Its strains were discovered worldwide, typically isolated from the samples of flowers and bumblebee honey [22]. According to Lachance et al., yeast strains of *Starmerella bombicola* were originally discovered in concentrated grape juice in South Africa, suggesting that nectar-feeding insects were most likely their dominant habitat of survival. Hence, yeast strain *S. bombicola* has high nutritional value because of their natural habitats of insects and plants [23]. Yeast *S. bombicola* can be defined as a species with potential to produce sophorolipids and cause no harm to human beings [22]. Due to alkane utilizing properties of yeast species, sophorolipids are mainly produced by *S. bombicola* with the use of first- and second-generation substrates. Solaiman et al. [24] reported that the utilization of second-generation substrates such as cheap waste stream substrates gave prominence to the production of sophorolipids in the recent decade, rather than the first-generation substrates such as vegetable oils and glucose [24]. Furthermore, in the study conducted by Cletus et al. [25], it was shown that sophorolipids can only be produced by *S. bombicola* subclade but not all species from the *Starmerella* clade can successfully synthesize sophorolipids. From this research, *S. bombicola* had the highest yield of sophorolipids after periods of incubation as compared with other *Candida* species such as *Candida kuoi* and *Candida cellae* [25].

Sophorolipids are glycolipid biosurfactants that are produced at relatively high yields by yeast strains [8]. Sophorolipids are glycolipids made up of a hydrophilic sophorose moiety coupled with a hydroxylated fatty acid tail of 16 or 18 carbon atoms and it is linked by a glycosidic bond [26]. The β-glycosidic bond is formed between anomeric carbon of the sophorose moiety (C1′) with the terminal or sub-terminal carbon of the hydroxylated fatty acids [8]. The carboxylic group of fatty acid can be in free (acidic) or internally esterified (lactonic) form (Figure 2).

#### 2.1.1. Acidic Compounds

Acidic forms of sophorolipids usually occur with free carboxyl groups of fatty acid tails linked to the sophorose moiety by a glycosidic bond. In general, acidic forms of sophorolipids tend to show better solubility as biosurfactants [27]. To explore antimicrobial potential of acidic sophorolipids, Lydon et al. [28] conducted research to evaluate antimicrobial activity of acidic sophorolipids in synthesizing pharmaceutical products to be used in wound healing [28]. The study showed that acidic sophorolipids have antimicrobial activity in both in vivo and in vitro experimental models, with no signs of adverse effects or inflammations [28]. Thus, acidic sophorolipids could be recommended as antimicrobial agents to ease the process of wound healing and minimize the risk of wound infection. 

#### 2.1.2. Lactonic Compounds

Lactonic sophorolipids are made up of a sophorose moiety attached to a long chain of hydroxyl fatty acids in which its carboxyl groups of fatty acids are esterified internally at the 4′- position of sophorose and 6′- or 6″- position in some rare cases, forming a closed ring structure [13]. Lactonic sophorolipids have lower water solubility than acidic sophorolipids due to the addition of acetyl groups into the compounds. Generally, sophorolipids in their lactonized form tend to display excellent surface activity and have great application potential [29]. Furthermore, the lactonic form of sophorolipids is reported to have higher antimicrobial activity compared with the acidic form, allowing for a better application in a broad range of industries [29]. Based on Shah et al. [30], the presence of acetyl groups in lactonic sophorolipids was found to have a huge difference on the potency and activity of the compounds [30]. From this research, acetylation of sophorolipids has been shown to reduce the hydrophilicity of the compounds and possess good spermicidal and anti-human immunodeficiency virus (HIV) activities [30]. Additionally, Structure Activity Relationship (SAR) analysis also indicated that esterification of carboxyl groups of fatty acids can influence the physicochemical properties and strength of the compounds [30]. 

### 2.2. Sophorolipids Biosynthesis

Sophorolipids production requires building blocks such as glucose and fatty acid. Both sources can be found in the production medium for the synthesis to take place [26]. Yeast strains such as *Candida bombicola* and *Candida apicola* have the essential enzymes to produce sophorolipids when growing on hydrophobic substrates such as alkanes, synthesizing fatty acids to be metabolized through the process of oxidation [26]. Hydroxylation of fatty acids by extracellular lipase will occur if there are fatty acids present in the culture medium. However, de novo synthesis of fatty acids from the acetyl-CoA produced by glycolysis will be done in the absence of fatty acids in the culture media [31]. Moreover, carbon sources such as a variety of sugars and oils can be used to synthesize sophorolipids effectively using *Candida bombicola* [32].

Sophorolipids are a glycolipid compound that is produced by a mixture of different molecules, which is commonly made up of glucose disaccharide sophorose and a hydroxylated fatty acid. The mixture of different congeners is susceptible to batch-to-batch variability of the compounds, resulting in discrepancy in the biological and physicochemical properties of the batches produced [33]. As such, in-product uniformity should be increased through production process and strain engineering to reduce batch-to-batch variation and ensure consistency. In contrast, the introduction of new types of natural producers can increase the inter-product variety of sophorolipids, which make them different from each batch [33]. Increasing attention on inter-product variation allows selection of compounds with differentiated configuration and performance to be applied in many diverse fields including food, pharmaceuticals, cosmetics, etcetera. 

To ensure inter-product variation in synthesizing sophorolipids, genetic modification and process engineering can be introduced to increase and enhance product variability and uniformity during the production process [33]. Based on Van Bogaert et al. [34], through the approach of genetic engineering, one of the yeast strains’, *Starmerella bombicola*, lactone esterase gene (also known as *sble*) was knocked-out while another strain acetyltransferase gene was knocked-out [34]. The product that is supposed to be observed is non-acetylated acidic sophorolipids, but interestingly, bolaform sophorolipids were obtained as well [34]. Bolaform glycolipid is made up of a hydrophobic fatty acid molecule with a sophorose head attached on both sides. Due to this unique structure, bolaform sophorolipid is expected to show unique physicochemical properties and activities compared to native sophorolipids, making them a great potential for applications in various aspects [35].

For example, bolaform sophorolipid has the potential to be applied in the biomedical field due to its outstanding features to control the interaction with cell surface receptors in moderation [35] and to provide tools for DNA purification [36]. The production of bolaform sophorolipids are usually obtained in very low amounts in the wild type sophorolipids mixture and process development strategies can be used to improve the production yield [33]. Based on Roelants et al. [37], the method of genetic engineering has been shown to provide new production of glycolipids with unique properties and gives a better understanding on the regulation of sophorolipid production [37]. Such new types of compounds offer more opportunities for the potential applications of biosurfactants in various fields.

#### 2.2.1. Biochemical Pathways

The general overview of the biochemical pathway in synthesizing sophorolipids is outlined in Figure 3. The first thing required to manufacture sophorolipids is the production medium of the hydrophobic and hydrophilic carbon sources such as fatty acids and glucose. In some cases, triglycerides, fatty acid methyl, or ethyl esters can also be used in the production medium [13]. From Figure 3, it is shown that the release of fatty acids is mediated via esterases. Firstly, the process of biosynthesis starts with the hydroxylation of hydrophobic carbon sources. Fatty acids must be hydroxylated by the cytochrome P450 (CYP450) monooxygenase from the CYP52 family and are converted to a terminal (ω) or subterminal (ω-1) position in fatty acids [13]. Hydroxylation of fatty acids takes place when the reduced nicotinamide adenine dinucleotide phosphate (NADPH)-dependent CYP450 monooxygenase enzyme is involved in the reaction [38]. The cytochrome P450 monooxygenase has high specificity to hydroxylate towards the ω-1 position in fatty acids [39].

Then, two molecules of UDP-glucose-dependent glycosyltransferases participate in the synthesis reaction, including UGTA1 and UGTB1 [40]. However, the glucose used for synthesis is produced during gluconeogenesis instead of just taken from the culture medium. Therefore, extra sugars are not required to be included in the culture medium [26]. One glucose molecule is attached to the hydroxyl group of the fatty acid at the C1′ position by a glycosidic bond through the glycosyltransferase I enzyme. Another glucose molecule is then linked to the first glucose moiety at the C2′ position by a glycosidic bond via glycosyltransferase II enzyme [13]. Subsequently, a nonacetylated acidic sophorolipid is obtained. 

In a gradual manner, the compounds can further undergo modifications such as acetylation of the sophorose group involving a specific enzyme named acetyltransferase [41]. Non-, mono- and diacetylated acidic forms of sophorolipids can be formed through the action of this enzyme by shifting an acetyl group from acetyl-coenzyme A (coA) to the sophorose moiety at the 6′- or 6″- position [40]. On the other hand, a hypothetical lactonizing enzyme found to be used in the lactonization of sophorolipids is lactone esterase. Lactonic sophorolipids are formed through esterification of the carboxyl group of fatty acids coupled with the hydroxyl group of the sophorose moiety. Esterification reactions often happen at the 4″- position of sophorose, and sometimes it takes place at the 6′- or 6″- position [13]. Based on Van Bogaert et al. [13], the ability of cellular lipases to develop ester linkage at the 6′- or 6″- position allows formation of lactonic sophorolipids without the action of lactone esterase. However, lactone esterase usually takes place at the 4″- position during esterification of carboxyl groups in the formation of lactonic sophorolipids [13].

Roelants et al. [37] reported that overexpression of the lactone esterase gene in sophorolipid production resulted in relatively higher amounts of products formed compared to wild type sophorolipids. Nearly all sophorolipids produced are lactonic form and approximately 3% of acidic sophorolipids will be produced when the gene is overexpressed. Vice versa, if the lactone esterase in the wild-type *S.bombicola* strain is knocked-out, only acidic sophorolipids will be formed but not lactonic sophorolipids under the similar conditions, which usually synthesize both types of sophorolipids in the production process [38].

#### 2.2.2. Regulatory Mechanisms

According to Lodens et al. [42], sophorolipid production can be regulated by the telomere positioning effect (TPE) in yeast species where the transcription of genes is suppressed when the sophorolipid gene cluster is positioned close to the telomere [42]. The presence of TPE is important in sophorolipid biosynthesis as it takes the key role to repress the gene clusters during unexpected exponential growth periods [42]. Additionally, the green fluorescent protein (GFP) reporter system has been used in this research to further analyze the effect of telomere positioning in gene expression of *S. bombicola* [42]. The introduction of the reporter system successfully supported the existence of TPE in the regulation of sophorolipid production [42]. 

As mentioned in Section 2.2.1, CYP52M1, UGTA1, UGTA2, and MDR proteins are required for extracellular transport of sophorolipids. Such genes are expected to be co-regulated as they have overlapping regulatory programs and have similar regulatory behaviors [43]. A study was conducted to evaluate the enzymes and other proteins that are involved in biosynthesis of sophorolipids [44]. The quantitative analysis was carried out using a lysine-deficient *S. bombicola* strain for sophorolipid production. By comparing the cell growth in early stationary phase (sophorolipid-producing) and exponential phase (sophorolipid non-producing), it was reported that in early stationary phase, high-level expression of sophorolipids gave rise to large amounts of products formed [44]. Therefore, the enzymes and proteins involved in the early stationary phase are far higher than those in the exponential phase [44]. 

### 2.3. Natural Roles

Sophorolipids produced by *Starmerella* clade were found to have many natural roles due to the positive correlation of yeast strains with flowers and honey of bumblebees [45]. Firstly, sophorolipids can enhance the absorption of hydrophobic substrates such as alkanes and triglycerides. For example, the utilization of octadecane in the culture medium can give rise to the production of sophorolipids, resulting in reduced energy use of yeast species in the production process [46]. Moreover, Hommel et al. proved that sophorolipids can function as an extracellular storage compound for energy and carbon [46]. According to Garcia-Ochoa and Casas [47], sophorolipids took the role of carbon source and were synthesized by yeast *Candida bombicola* to form sophorose disaccharide [47]. 

Furthermore, antimicrobial properties can be observed in different mixtures of sophorolipids to impede the growth of microorganisms such as bacteria and fungi. Based on Ito et al. [48], the lactonic form of sophorolipid produced by *S. bombicola* has antimicrobial properties such as microbial growth inhibition, as in *Candida albicans* [48]. The antimicrobial activity of sophorolipids mainly targets the destruction of the cellular membrane and interferes with the structure of microorganisms, resulting in growth inhibition of pathogenic microbes not limited to bacteria only, but also yeasts and fungi [49].

Surface-active properties of sophorolipids depend on the measure of surface tension and critical micelle concentration (CMC) of the compound [26]. The properties of surface and interfacial tension of sophorolipids are based on their configuration and structures with different degrees of acetylation. Sophorolipids have been reported to lower the surface tension of water while dissolving in aqueous solution [13]. Additionally, a low CMC value indicates that low concentration of sophorolipids is used to decrease the surface and interfacial tension at the interface [13]. Hence, sophorolipids possess good emulsifying and solubilizing properties, making them a good candidate in manufacturing cleaning agents, wetting agents, and stabilizers [13]. Acidic forms of sophorolipids commonly show better performance of foaming and solubility properties while the lactonic forms of sophorolipids generally possess a higher level of surface activity than the acidic form [13].

In addition, biological and physicochemical properties can be affected due to different forms and structures of sophorolipids. For instance, esterified sophorolipids such as sophorolipid-hexyl ester exhibited superior emulsifying properties on the oil–water system compared to the non-ionic surfactants such as Triton X-100 [50]. According to Ma et al. [51], surface and functional activities of both forms of sophorolipids, lactonic and acidic, could be influenced by the length of the carbon chain as well as the degree of saturation and acetylation of the compounds [51]. The emulsifying capacity and surface activity of sophorolipids will increase when the length of the carbon chain in the structure is increased [52]. Lactonic sophorolipids have been reported to show higher CMC values than acidic sophorolipids [51]. Moreover, another study showed that natural sophorolipids containing a lactic-acid ratio of 72:28 exhibited the lowest surface tension and CMC value compared to the other sophorolipids tested due to the increased hydrophobicity of lactonic compounds [53]. Additionally, lactonic sophorolipid appeared to have higher cytotoxic effects relative to the sophorolipid without lactonic properties [51].

## 3. Methodology

### 3.1. Literature Search

A well-organized search was conducted to identify all relevant published literature for good quality references on the applications of sophorolipids in food and health industries. An in-depth literature search could help to identify the research gap and application trends by evaluating the available articles gathered. 

### 3.2. Search Terms and Databases

The literature search identifying studies of applications of sophorolipids in food and health industries was conducted using keywords to include the relevant research articles: “sophorolipid” OR “sophorolipids” OR “biosurfactant” OR “biosurfactants” OR “glycolipid biosurfactant” OR “glycolipid biosurfactants” AND “application” OR “applications” OR “uses” OR “utilization” OR “utilisation”. This was performed by using research databases including PubMed, ScienceDirect, and EBSCOhost for the collection of research articles. Manual searches from reference lists of the screened articles were also conducted to identify additional primary studies. 

### 3.3. Inclusion and Exclusion Criteria

This review included studies with experiments conducted in vivo and in vitro for applications of sophorolipids in food and health industries, articles published in the English language from 2011 to 2022, and articles that their full texts are available. However, research articles published in a language other than English, studies that were published before 2011, articles without full text, and duplicated studies were excluded from this review. 

### 3.4. Study Selection

All potential research articles that met the inclusion and exclusion criteria were selected by screening the titles and abstracts. Full texts of the qualified papers were retrieved and screened for eligibility based on the inclusion criteria. 

### 3.5. Data Extraction 

The author determined the suitability of the studies for inclusion in the review by reviewing the titles and abstracts of the selected papers. Data extraction from each identified study was performed by the author and recorded into a data extraction table containing detailed information about the structure and bioactivity of the compounds and their corresponding applications.

## 4. Applications of Sophorolipids in Food and Health

### 4.1. Food

Sophorolipids are used in many food applications including food preservation, agricultural practices, and bioconversion from food waste. They can be applied as a form of strategy for food waste management for sustainable production of sophorolipids.

#### 4.1.1. Food Preservation

Olanya et al. proposed that pathogen control by sophorolipids is important to provide health benefits for consumers [54]. In recent years, foodborne pathogens continue to rise and thus result in significant food contamination and food safety risks to consumers throughout the world [55]. The most common types of pathogens that present in food causing foodborne illness include *Salmonella enterica* serovars, *Escherichia coli* O157:H7, and *Listeria monocytogenes* [56]. Therefore, control of foodborne pathogens is of immediate concern in the population to provide food and health safety for consumers. 

Biobased antimicrobial compounds such as sophorolipids have been reported to show beneficial effects for pathogen control due to their properties of having a low toxicity and being highly biodegradable [57]. These properties demonstrated by sophorolipids can help in enhancing postharvest processes to produce fresh and safe fruits and vegetables for consumers [57]. Additionally, the combined usage of sophorolipids with sanitizer was shown to give synergistic antimicrobial effects on *E. coli* O157:H7. However, limited data were documented for the application of pathogen control by combining sophorolipids with antimicrobial sanitizer [54].

Sophorolipids were also reported to exhibit good performance in growth inhibition of foodborne fungi such as *F. oxysporum* through membrane destabilization of fungal pathogens [58,59,60]. Such sophorolipids could be produced by *S. bombicola* [58], *Metschnikowia churdharensis* [59], and *Rhodotorula babjevae* YS3 [60].

Sophorolipids have been introduced as a useful material in manufacturing food packaging films and are widely known to display antimicrobial activity directly against the pathogenic microorganisms found in food [61]. According to Silveira et al. [61], antimicrobial packaging films were made by using polylactic acid (PLA) and sophorolipids, which were used as film plasticizer via a casting process in order to control foodborne pathogens [61]. Results showed that good dispersion of the compounds is achieved and forms a smooth appearance of the films, hence the films produced are beneficial in food packaging because a good appearance of the package is the main concern for consumers [61]. Additionally, the addition of sophorolipids into PLA films make them more water-soluble due to the hydrophilic sugar moiety of sophorolipids [62,63]. Higher water solubility of the films contributes to a better release of the antimicrobial components in the food itself, providing antimicrobial effects against the poultry pathogens such as *Staphylococcus aureus*, *Salmonella* spp., and *Listeria monocytogenes* [64].

Further research reported on the different structures and composition of sophorolipids for the antimicrobial efficiency against foodborne pathogen *Escherichia coli* O157:H7 [65]. As stated by Zhang et al. [65], lactonic sophorolipids from *S. bombicola* showed a significant decline in *E. coli* population growth compared to the corresponding acidic sophorolipids. Moreover, the introduction of ethanol into sophorolipids was studied and was shown to provide increased antimicrobial effects to the sophorolipids for growth inhibition of pathogens by interrupting the membrane integrity of the cells [65,66]. Zhang et al. [67] observed that lactonic sophorolipids derived from oleic, palmitic, and stearic acids were more effective than acidic forms against *Listeria* spp. (Gram-positive bacteria) and *Salmonella* spp. (Gram-negative bacteria), in which *Listeria* spp. showed a higher sensitivity towards sophorolipids [67]. 

Production of sophorolipids from marine yeast *Rhodotorula rubra* has been observed to show antibacterial activities against foodborne pathogens *E. coli* and *S. aureus* with the utilization of *Macrocystis pyrifera* extract as a nutrient source [68]. Similar antibacterial potentials were demonstrated by Chen et al. when they combined nisin with sophorolipids to form antimicrobial food preservatives [69]. Based on Gaur et al. [70], sophorolipids from *Candida albicans* and *Candida glabrata* have been shown to be effective antibacterial biomolecules against both Gram-positive and Gram-negative bacterial pathogens through the formation of reactive oxygen species (ROS) [70]. 

The emulsifying capacity of sophorolipids has been evaluated against different vegetable oils and results showed that sophorolipids from *Candida* species have high emulsifying activity, developing great promise for their use in the food industry [70]. Sophorolipids were reported to be excellent emulsifying agents for various oil phases such as lemon oil [71] and oregano oil [72] to enhance the stability and texture of food products [73].

Silveira et al. [74] utilized poultry isolates of *Clostridium perfringens* and *Campylobacter jejuni* cultures to evaluate the antibacterial activity of sophorolipids from *S. bombicola* [74]. In this study, the combination of sophorolipids with lactic acid showed the additive effects of antibacterial properties against the pathogens mentioned [74]. Years later, lactonic sophorolipids and lactic acid were studied again for their synergism against *L. monocytogenes* and *S. aureus*, which are different types of poultry pathogens compared to the previous study [75]. This combination was effective for growth inhibition of the pathogens due to their known antibacterial properties [75].

#### 4.1.2. Agricultural

Few studies support antimicrobial action of sophorolipids on phytopathogenic microorganisms. Sophorolipids were reported to have inhibitory effects on plant pathogens such as *Pythium ultimum* [76] and *Moesziomyces* sp. [77], making them suitable antimicrobial agents against plant pathogens in the agriculture industry. Additionally, sophorolipids from *Wickerhamiella domercqiae* were tested against fungal and oomycete pathogens such as *Fusarium oxysporum* and *P. ultimum* and showed that they inhibited mycelial growth and spore germination of pathogens [78]. A study performed by Vaughn et al. [78] reported that sophorolipids were found to be applied as postemergence herbicides for enhanced plant cuticle permeability [79]. 

#### 4.1.3. Bioconversion from Food Waste

Bioconversion of food waste into sophorolipids has been reported to show potential in a broad range of applications. Inedible Jatropha oil can be used as fermentation feedstock in the production of sophorolipids by *S. bombicola* [80]. The synthesized sophorolipids have excellent surfactant properties and can reduce tension of distilled water with a low critical micelle concentration (CMC) value [81]. Joshi-Navare et al. [81] demonstrated that oil-derived sophorolipids displayed better emulsification activity and stability as compared with those standard chemical surfactants based on the evaluation of environmental parameters such as pH, temperature, and water hardness [81]. 

Waste glycerol can be used as fermentation feedstock to produce sophorolipids by various yeast species such as *Candida floricola* [82]. From the study, it was reported that yeast strain *C. floricola* with the use of waste glycerol as a carbon source can produce only the acidic form of sophorolipids but not lactonic sophorolipids [82]. This is because the absence of the lactone esterase enzyme for esterification of the compound will result in the production of sophorolipids without lactone forms by the strain of *C. floricola*. Hence, waste glycerol serves as a cost-effective feedstock for sophorolipid production which suggests its potential use in various aspects [82]. Another study presented by Wang et al. [83] showed that waste streams can also be utilized as feedstock to produce sophorolipids from *S. bombicola* via fed-batch fermentation with a volumetric productivity of 3.7 g/L/h [83]. 

Sunflower acid oil, which is a by-product from vegetable oil refineries, was evaluated as fermentation feedstock to produce both acidic and lactonic sophorolipids from *S. bombicola* [5]. Sophorolipids formed were shown to have good surface activity, wetting and foaming capacity, as well as superior emulsifying activities [5]. Residual sunflower oil cake obtained from the oil industry through the process of winterizing was proven successful in the production of sophorolipids via solid-state fermentation by *S. bombicola* [84]. Good emulsifying properties and displacement activities were observed from the sophorolipids formed by waste from the oil processing industry [84].

Successful industrial production of sophorolipids can be obtained from the bioconversion of food waste through the process of enzymatic hydrolysis [85] and conversion of waste frying oils using dual lipophilic substrates [86]. Kim et al. [39] further investigated the production of sophorolipids by valorization of waste cooking oils through the process of fed-batch fermentation of yeast *S. bombicola* [39]. Later, sophorolipids formed by the bioconversion of waste cooking oils were further explored for bioplastics production by synthesizing methyl hydroxy branched fatty acids (MHBFAs) as co-monomers [39]. Today, bioplastics have been of interest to the public as they are recognized as lower toxicity products and are better for the environment compared to traditional plastics [87]. The use of sophorolipids in the food industry is summarized in Table 1.

### 4.2. Health

#### 4.2.1. Cosmetic Formulations

Sophorolipids derived from hydrolyzed horse oil have been proven to show excellent anti-wrinkle effects as well as to improve skin elasticity and firmness [88]. Additionally, sophorolipids have shown antibacterial effects against *Pseudomonas aeruginosa*, *Staphylococcus aureus*, and *Escherichia coli* as well as displayed antifungal activity against *Candida albicans* and *Aspergillus niger*, in which all five of these microbes are recognized as the most relevant microorganisms found in cosmetic formulations [88]. Based on Zerhusen et al. [89], the formation of long chain non-ionic sophorolipids was reported to lower the surface tension between phases and to exhibit potent emulsifying activities in oil-water mixtures [89]. The produced sophorolipids had good emulsifying behavior as they stabilized the emulsion and prevented water and oil phases from separating, making them superior in manufacturing of pharmaceutical creams, ointments and lotions [89]. 

A few studies highlighted the use of sophorolipids as natural antimicrobial agents for applications in skincare pharmaceutical formulations due to their non-toxic nature and good skin compatibility [90,91,92]. The authors in that study observed antimicrobial efficiency on *Propionibacterium acnes* by sophorolipids embedded in different composite films such as plant-based composites pectin and alginate [90] and poly(3-hydroxybutyric acid-co-3-hydroxyvaleric acid) (PHB/HV) composites [91]. Solaiman et al. [92] demonstrated that long carbon chains of acidic sophorolipids (22 carbon chains) had the strongest antibacterial effects on the tested bacteria *Cutibacterium acne* compared to the others including the lactonic form of sophorolipids consisting of 18 carbon chains only [92]. These results suggest that 22:0 sophorolipids are better suited biosurfactants as they possess antimicrobial properties for various applications [92]. 

Sophorolipids, which often form themselves into a vesicle-like structure through self-assembling, are known to be effective in skin delivery of molecules for application in cosmetic and pharmaceutical industries. Based on Ishii et al. [93], acidic sophorolipids play a role in increasing skin permeation and achieving higher amounts of bovine lactoferrin penetration across the skin barrier [93]. This study highlighted the use of sophorolipids as a suitable carrier for transdermal delivery because they do not affect the original property of lactoferrin but to improve the skin absorption of lactoferrin [93,94]. Furthermore, formation of biodegradable transferosomal hydrogels for cosmetic applications has been investigated by Naik et al. via the combined use of lignans and sophorolipids [94]. Sophorolipid-based transfersomes were proven to enhance skin permeation for transdermal delivery of active components across the skin barrier due to their amphiphilic properties [94]. Imura et al. [95] also examined the transdermal permeation and absorption of mogrosides V by incorporating triterpene glycoside into the vesicles of acidic sophorolipids to display various pharmacological activities of mogrosides V [95].

#### 4.2.2. Wound Healing

Sophorolipids were shown to be effective for growth inhibition of bacterial pathogens when combined with antibiotics such as kanamycin or cefotaxime [28] as well as natural compounds such as sericin and calcium alginate [96]. These results suggested the potential use of sophorolipids as an active ingredient in antimicrobial formulations for wound healing applications, without any evidence of side effects on skin tissue [28]. This combination demonstrated a rapid rate of wound healing and contraction, thus shorter time is used for the wound healing process compared to the traditional formulation [96]. The healing potential of sophorolipids was demonstrated using HT-29 cell lines and results showed that sophorolipids could increase cell proliferation and migration, which is beneficial in the application of intestinal healing [97]. 

#### 4.2.3. Antimicrobial: Antifungal, Antibacterial, Biofilm Destruction

Sophorolipids are widely studied in industrial and pharmaceutical industries due to their possession of unique properties, being antiviral, antibacterial, antimicrobial, and antibiofilm. The potential use of sophorolipids as antimicrobial formulating agents has been of great interest in recent years due to their attractive properties such as low toxicities and better biodegradability. Their ability to show numerous biological activities makes them suitable and efficacious alternatives for synthetic surfactants in the pharmaceutical sectors [98,99].

According to Díaz De Rienzo et al. [57], sophorolipids from *C. bombicola* were shown to have antibacterial properties against both Gram-positive and Gram-negative microorganisms by inducing plasma membrane damage of the microbes [58]. Cell membrane disruption occurs when sophorolipids can alter the morphology and structure of the bacteria resulting in increased membrane permeability and disturbance of membrane integrity [100]. This ensures sophorolipids penetrate cell membranes and release intracellular materials, causing ruptures in cell membranes and growth inhibition of the bacteria [101]. In general, it might be somewhat difficult for sophorolipids to penetrate the cell membrane of Gram-negative bacteria due to the presence of lipopolysaccharide surrounding the outer cell membrane [102]. However, this study showed that sophorolipids inhibit the growth of both Gram-positive and Gram-negative bacteria at the same MIC of 5% *v/v* [100]. Other similar studies also demonstrated antibacterial activity of sophorolipids produced from *Candida* species such as *C. bombicola* and *C. tropicalis* RA1 against Gram-positive bacteria [75,103,104]. Meanwhile, Archana et al. [104] also reported on the antibacterial efficiency and the growth inhibition of Gram-negative bacteria such as *E. coli* and *P. aeruginosa* [104]. Oil-derived sophorolipids, which are natural products, could be used to replace synthetic surfactant detergent formulations as they were reported to display antibacterial activity against *S. aureus* [81]. Abhyankar et al. [105] also studied the antibacterial activity exhibited by myristic acid derived sophorolipid against both Gram-positive and Gram-negative organisms [105]. 

On top of that, the antibiofilm and antiadhesive potential of sophorolipids were observed against Gram-positive bacteria [100]. Such action is achieved through alteration in surface properties of the bacterial cells and antiadhesive activities exhibited by sophorolipids [100,106]. Valotteau et al. [107] reported the ability of sophorolipids to disrupt biofilm formation and reduce bacterial adhesion of pathogen strains such as *S. aureus* and *E. coli* [107]. The action of sophorolipids on the inhibition of biofilm formation and reduced microbial adhesion from different surfaces could suggest their promising use in various industries including the biomedical and pharmaceutical sectors [107]. Moreover, sophorolipids had been proven to display antibiofilm properties and prevent cell attachment of *S. aureus*, suggesting them to be coating materials in medical-grade silicon devices for application in the pharmaceutical industry [108,109].

Nguyen et al. [110] also proved the antibiofilm activity of sophorolipids combined with sodium dodecyl sulfate (SDS), which is an anionic surfactant, against *Pseudomonas aeruginosa* PAO1 [110]. However, they found that sophorolipids do not show any antibacterial action on PAO1 but only disperse biofilm formation of the bacterial strain [110]. Vasudevan and Prabhune [111] evaluated that curcumin-sophorolipid nanoparticles exhibited good antibiofilm activities by quorum quenching against *P. aeruginosa* [111]. 

Sophorolipids were reported to have antifungal action against *C. albicans* by interrupting their growth and the formation of biofilm of the fungal strain [7,112,113]. Such actions were also achieved through increased cell permeability and generation of reactive oxygen species (ROS), resulting in fungal necrosis and apoptosis due to high oxidative stress [7]. Furthermore, sophorolipids were found to prevent fungal infections such as tinea pedis and dermatophytosis because of their evident antifungal activities against *Trichophyton mentagrophytes* [114,115].

Dengle-Pulate et al. [116] also examined sophorolipids synthesized from medium-chain lauryl alcohol for their antibacterial effects on various pathogenic microorganisms and revealed that lauryl alcohol derived sophorolipids (SLLA) exhibited higher antibacterial activities than lauryl alcohol alone [116]. A lower concentration of SLLA was utilized to show their antibacterial effect against Gram-positive *S. aureus* while SLLA showed complete inhibition towards Gram-negative *E. coli* at a higher concentration [116]. Interestingly, they found that SLLA possesses excellent antifungal activities against *C. albicans* at a concentration of 25 μg/mL and reported that *C. albicans* is fully inactivated at a higher concentration of 50 μg/mL [116]. These outstanding properties make sophorolipids potent to be used as antibacterial or antifungal agents in biomedical and therapeutic applications. 

Additionally, sophorolipids were reported to effectively exert antibacterial activities against microorganisms when they are coated on gold surfaces [99]. From this study, the sophorolipid monolayer was chemically attached on a gold substrate and was proven to show antibacterial effects against non-pathogenic *Listeria ivanovii* and several pathogens such as *Streptococcus pyogenes* and *E. coli* at a concentration of 5 μg/mL [99]. Due to the attractive antimicrobial properties towards several bacterial strains, sophorolipids-grafted surfaces are gaining more attention in pharmaceutical applications [99].

Antimicrobial action against oral pathogens such as *Streptococcus oralis* [117] and *Lactobacillus acidophilus* [118] have been reported through inhibition of biofilm formation of oral cariogenic bacteria. Solaiman et al. [119] studied the antimicrobial activity of sophorolipids against a mixed culture of Gram-positive and Gram-negative bacteria and revealed that sophorolipids were effective against a wide range of microorganisms found in hides, which will be useful for further application in the leather industry [119]. 

Joshi-Navare and Prabhune [120] further proved that sophorolipids have potent antimicrobial activity against bacterium *E. coli* and *S. aureus* in combination with antibiotics [120]. However, growth inhibition of bacteria was not completely performed by sophorolipids alone [120]. In this study, sophorolipids combined with tetracycline inhibited growth of *S. aureus* while the antibacterial action against *E. coli* was observed through the combination of sophorolipids with cefaclor [120]. Combinatorial antibacterial effects of antibiotics and sophorolipids have been shown to exhibit adjuvant activities against bacterial pathogens as well as to overcome the issue of bacterial antibiotic resistance [28,120]. In addition, incorporation of acidic sophorolipids into amphotericin B was reported to show antifungal and antibiofilm effects against *C. albicans* in the treatment of fungal infections. This study indicated that sophorolipids can be used to develop fungicidal agents with amphotericin B by preventing gene expression and growth of fungal pathogens [113]. 

Baccile et al. [121] reported that acidic sophorolipids from *C. bombicola* have been used to develop functionalized iron oxide nanoparticles due to great colloidal stability of the compound [121]. A similar study also showed that sophorolipid was found to be a good stabilizer in forming zinc oxide nanoparticles (ZON) to show inhibitory effects on the tested microorganisms including *Salmonella enterica* and *C. albicans* [122]. They reported that antimicrobial activities were exhibited by diacetate acidic sophorolipids from *Cryptococcus* sp. against both bacterial and fungal pathogens [122]. Owing to the antimicrobial trait of sophorolipids and their additive effects with zinc oxide, they can be explored in the production of functionalized nanoparticles for the control of pathogenic microbes [122]. Another study showed that long-chain quaternary ammonium sophorolipids possess antimicrobial activities towards both Gram-positive and Gram-negative bacteria, in which a higher MIC of sophorolipids is used for growth inhibition of Gram-negative *E. coli* [123]. 

#### 4.2.4. Anticancer

The anticancer effects of sophorolipids with different structures were reported by a few researchers, in which C18:1 DLSL was shown to have the highest activity compared to diacetylated lactonic sophorolipid with a C18 saturated fatty acid (C18:0 DLSL) against human esophageal cancer cells [124], breast cancer cells [17], human cervical cancer cells [125], and colorectal cancer cells [126]. These results indicated that increasing the degree of unsaturation of the compound will result in lower efficiency on apoptosis of cancer cells. 

Sophorolipids combined with cetyl alcohol (SLCA) were shown to have anticancer activity against human cervical cancer cells by inducing apoptosis through a rise in intracellular calcium ions leading to the depolarization of mitochondrial membrane potential [127]. Anticancer action against colon cancer cell lines by sophorolipid-based nanocapsules was demonstrated by Haggag et al. in both in vivo and in vitro experiments [128]. Lactonic sophorolipids were shown to be effective in growth inhibition of liver hepatocellular carcinoma cells [129] and inducing angiogenesis [130]. Sophorolipids have been shown to target the cancerous cells without affecting the normal cells, thus reducing unwanted side effects that are normally associated with the current therapeutic regimens [127]. The use of sophorolipids in health is summarized in Table 2.

## 5. Future Perspectives

Through analysis of the publication activities in various research databases, the use of sophorolipids in different areas has been on a rising trend. Therefore, sophorolipids appear to be promising compounds that can be used in substituting the usage of chemical surfactants in different industries due to being Earth-friendly and safe for human use. Further research could, for instance, explore the production of different types of structures and composition of sophorolipids, while maintaining their non-toxic and environmentally friendly properties. Such research could contribute to extending their functionalities and biological activities related to potential applications in a wide range of sectors. 

The present review focused on the potential roles of yeast strains isolated from flowering plants and honeybees in the production of sophorolipids. Further research should be undertaken to investigate the isolation of new types of microbial strains that can rapidly produce sophorolipids under different culture conditions. We believe that different structures of bioactive sophorolipids can be identified depending on their production, especially from the unexplored environments. Additionally, strain improvement should be studied to increase the productivity of sophorolipids through different methods such as genetic recombination and mutation. This is an issue for future research to explore and enhance bio-extraction of sophorolipids from existing fungal isolates and novel bacterial strains. 

In future investigations, nanotechnology can be used in the application of sophorolipids in the food sector, which has not been studied in the past 10 years. For instance, it might be possible to perform the incorporation of sophorolipids in the production of nano formulations for evaluation of its encapsulation and release efficiency in food processing, including enhancement of nutritional value and shelf-life extension of food products. Furthermore, the present results on biofilm inhibition and wound healing applications are rather disappointing. Future research on biofilm dispersal of sophorolipids and their associated antimicrobial and antifungal mechanisms are recommended for the development of antimicrobial agents with more efficient roles in growth inhibition of multiple bacteria. More investigations of wound healing activity should be performed to further support the incorporation of sophorolipids into a safe and eco-friendly wound dressing. 

## 6. Conclusions

Sophorolipids are an attractive alternative in the agricultural and industrial market nowadays as they are synthesized from natural sources and have several advantages compared to synthetic surfactants. It is one of the important questions that many researchers are investigating to provide social and environmental safety in industries such as food, agriculture, pharmaceutical, cosmetic, petroleum, and many others. This review paper highlights the fundamental principles of sophorolipids and existing published articles on their application in various sectors. Sophorolipids have gained considerable attention in versatile fields over the years due to their diversity of structures and the production from low-cost fermentation substrates. The current published data highlights the importance of sophorolipids as natural compounds for potential application in a wide range of sectors through sustainable technologies. The review paper contributes to our understanding of sophorolipids as biosurfactants with countless advantages to humans and the environment. However, the scope of this data was limited in terms of the years of published data, languages used and the accessibility of the full-text articles. Despite its limitations, the study certainly adds to our understanding of the basic concept of sophorolipids and their promising applications in food and health industries. More broadly, research is needed to involve advanced technologies in the production and applications of sophorolipids, as well as focus on exploring how sophorolipids can be produced efficiently based on different production processes. Overall, the present study identifies the research gap and provides further insights for future research.

## Figures and Tables

**Figure 1 molecules-27-05556-f001:**
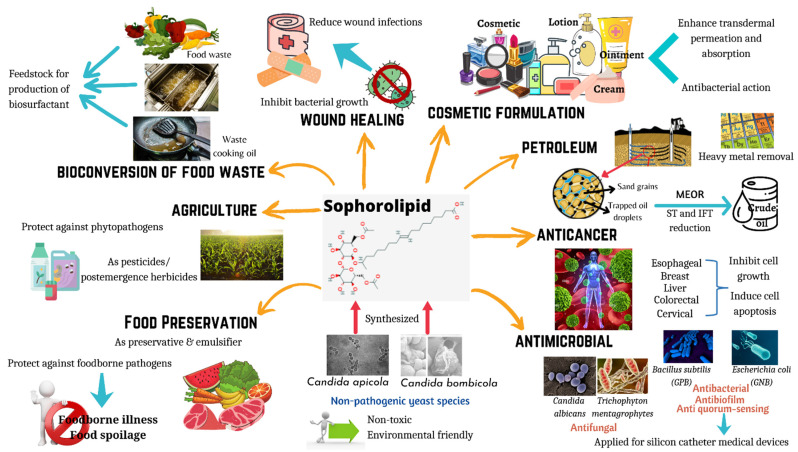
Applications of sophorolipids in food and health. Sophorolipids are synthesized by non-pathogenic yeast species, and they have the potential to be used in wide range of applications including wound healing, cosmetic formulations, anticancer agents, antimicrobial agents, food preservation, food waste management, and agriculture practices.

**Figure 2 molecules-27-05556-f002:**
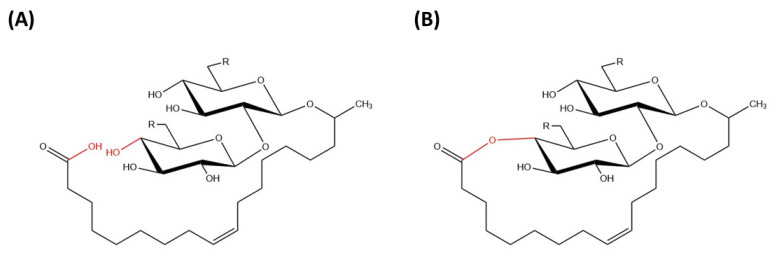
Chemical structure of (**A**) acidic sophorolipid (**B**) lactonic sophorolipid.

**Figure 3 molecules-27-05556-f003:**
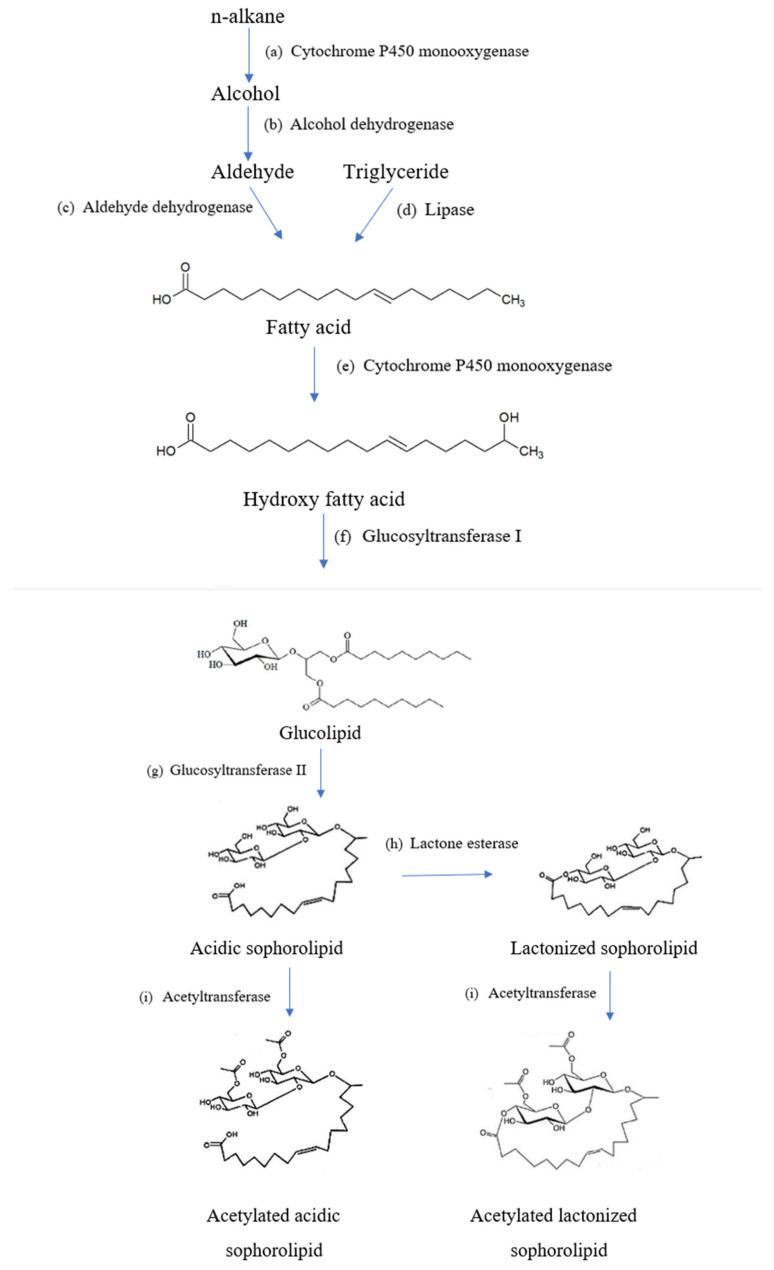
Overview of biochemical pathways in sophorolipid production.

**Table 1 molecules-27-05556-t001:** Application of sophorolipids in the food industry.

Applications	Source of Sophorolipids	Structure	Bioactivity	Concentration (Types of Microbes)	Mechanism(s)	Ref.
Food preservation	*Starmerella* *bombicola*	Lactonic	Antimicrobial	1% (*E. coli* O157:H7)	Protection against foodborne pathogenic bacteria	[65]
			Antibacterial; Emulsifier; Preservative	MIC ^1^ 32 µg/mL *(S. aureus)*	Protection against foodborne pathogens; serve as an additive with nisin	[69]
			Antifungal	MIC/MFC ^2^ 729.0 µg/mL *(Fusarium oxysporum)*	Protection against foodborne pathogens/food spoilage fungi	[58]
			Antibacterial	MIC 0.0015% *(Clostridium perfringens)*, 0.5% *(Campylobacter jejuni)*	Protection against food pathogens in poultry industry	[74]
				MIC 31.25 µg/mL *(S. aureus)* 62.5 µg/mL *(L.monocytogenes*)	Protection against food pathogens in poultry industry	[75]
			Emulsifier	0.1 wt%	Improve the stability of oil-in-water emulsions	[73]
		Acidic and lactonic	Emulsifier; reduce interfacial tension	1 wt%	Improve the stability of lemon oil-in-water emulsions	[71]
		Lactonic Sophorolipid	Emulsifying properties	0.5 wt%	Stabilize oregano oil-in-water emulsions	[72]
	*Candida albicans* and *Candida glabrata*	Sophorolipid	Emulsifier, Antibacterial	60 µg/mL (*B. subtilis*)	Protection against bacterial pathogens, possible use as food emulsions	[70]
	*Rhodotorula rubra*	Acidic	Antibacterial	200 µg/mL *(E. coli and S. aureus)*	Protection against foodborne pathogens	[68]
	*Rhodotorula babjevae YS3*	Acidic; lactonic	Antifungal	MIC 125 µg/mL *(Fusarium oxysporum)*	Protection against pathogenic fungi	[60]
	*Metschnikowia churdharensis*	Acidic; lactonic	Antifungal	MIC 49 µg/mL *(F. oxysporum)* 98 µg/mL *(F. solani)*	Protection against food spoilage fungal pathogens	[59]
Agricultural	*Starmerella* *bombicola*	Lactonic	Antimicrobial	MIC 2 mg/mL *(Pythium ultimum)*	Protection against phytopathogens/conventional pesticides for tomato plants and fruits	[76]
			Antifungal	1 mg/mL *(Moesziomyces* sp.)	Protection against plant pathogen	[77]
	*Wickerhamiella domercqiae*	Lactonic	Antimicrobial	10 mg/mL *(F.oxysporum,* *P. ultimum)*	Inhibition of spore germination and mycelial growth of pathogens	[78]
	*Candida kuoi*	Acidic	Emulsifying properties; herbicidal activity	1% *v/v* *(Senna obtusifolia)*	Phytotoxicity against sicklepod; Used as postemergence herbicides	[79]
Bioconversion from food waste	*Starmerella* *bombicola*	Acidic and lactonic	Emulsifier	51.5 g/L (submerged fermentation)	Sunflower oil refinery waste as feedstock	[5]
			Surface-active property; Antibacterial	15.25 g/L (resting cell method) CMC ^3^: 9.5 mg/L MIC90 ^4^: 300 μg/mL (*S. aureus*)	Non-edible Jatropha oil as feedstock; Replace synthetic surfactants in detergent	[81]
		Lactonic	N/A	3.7 g/L (fed-batch fermentation)	Waste stream and food waste as feedstock	[83]
			N/A	115.2 g/L (batch fermentation)	Bioconversion of food waste by enzymatic hydrolysis	[85]
		C18:1 DLSL	Emulsifying properties	1 g/L (solid-state fermentation) CMC: 40.1 mg/L	Winterization oil cake as feedstock, used in diesel displacement	[84]
	*Candida floricola*	Acidic	N/A	3.5 g/L (fermentation process using glycerol)	Waste glycerol as fermentation feedstock	[82]

^1^ MIC: Minimum inhibitory concentration; ^2^ MFC: Minimum fungicidal concentration; ^3^ CMC: Critical Micelle Concentration; ^4^ MIC_90_: Minimum Inhibitory Concentration required to inhibit the growth of 90% of organisms. N/A: Not applicable.

**Table 2 molecules-27-05556-t002:** Applications of sophorolipids in health.

Applications	Source of Sophorolipids	Structure	Bioactivity	Concentration (types of Microbes)	Mechanism(s)	Ref.
Cosmetic and wound healing	*Starmerella* *bombicola*	Acidic (from *C. kuoi*); Lactonic (from *S. bombicola*)	Emulsifier	50 μg/mL	Cosmetic creams and lotions, pharmaceutical ointments	[89]
		Acidic and lactonic	Antimicrobial	0.24% *w/w* *(Propionibacterium* *acnes)*	Anti-acne agent	[90]
		Acidic	Enhance transdermal absorption of lactoferrin	0.01%	Dermal fibroblast proliferation (cosmetic use)	[93]
		Sophorolipid (with lignans)	Transdermal permeation	10 μg/mL	Design biodegradable transferosomal hydrogel (cosmetic use)	[94]
		Acidic	Reduce surface tension	CAC ^2^ 0.083%	Skin penetration enhancer	[95]
		Acidic (C_18”1_-NASL) ^3^	Antimicrobial	MIC 4 mg/mL *(Enterococcus faecalis*, *P. aeruginosa)*	Applied with adjuvant antibiotics (kanamycin or cefotaxime) in wound healing	[28]
		Sophorolipid-Sericin Gel	Antibacterial	500 μg/mL (76.1% antioxidant activity)	Wound healing in wistar rats	[86]
	*Rhodotorula* *bogoriensis*	C_22_-SL ^1^	Antimicrobial	100 mg/mL (*Propionibacterium* *acnes)*	Inhibit growth of *P. acne* (skin acne)	[91]
	*Pseudohyphozyma bogoriensis*	6′-Ac-22:0-SL (22 carbon chains) (acidic)	Antibacterial	CMC 10 μg/mL *(Cutibacterium acne)*	Inhibit growth of *C. acne* (skin acne)	[92]
	N/A	Diacetylated lactonic sophorolipid	Immunomodulatory properties	25 μg/mL (suppressed M1 macrophages polarization)	Used as coatings to promote the resolution of inflammation and normal wound healing	[98]
	N/A	Sophorolipid	Antimicrobial	10 μg/mL (*E. coli*, *Streptococcus* spp. and *Salmonella* spp.)	Accelerate proliferation and migration in vitro wound model using HT-29 cells, accelerate intestinal wound healing	[97]
Antimicrobial	*Starmerella* *bombicola*	Acidic and lactonic	Antibacterial Antibiofilm	MIC >5% *v/v* *(Cupriavidus Necator, Bacillus subtilis)*	Induce cell death in planktonic cells	[57]
				MIC 2 mg/mL *(B. subtilis)* 1 mg/mL *(S. aureus)* 4 mg/mL *(E. coli, P. aeruginosa)*	Inhibit both GPB ^4^ and GNB ^5^	[104]
				MIC 150 μg/mL *(S. aureus*) 350 μg/mL *(P. aeruginosa)*		[105]
				MIC_90_ 300 μg/mL *(S. aureus)*	Inhibit GPB	[81]
				200 μg/mL *(B. subtilis)*		[75]
		Nonacetylated acidic	Antibacterial	5 μg/mL *(Listeria ivanovii)*	Inhibit GPB	[99]
		Lactonic	Antibacterial	MIC 400 μg/mL (*S. aureus, E. coli,)*	Inhibit both GPB and GNB in synergy with antibiotics	[120]
		Acidic and lactonic	Antimicrobial Antifungal	MIC 6 μg/mL (*S.aureus*) 30 μg/mL (*E. coli*) 50 μg/mL (*C. albicans*)	Inhibit bacterial and fungal infections	[116]
		Lactonic	Antifungal	MIC_80_ ^6^ 60 μg/mL (*C. albicans*) BIC_80_ ^7^ 120 μg/mL (*C. albicans*)	Inhibit biofilm formation and hyphal growth of *Candida* species	[112]
		Acidic	Antifungal Antibiofilm	MIC_70_ ^8^ 24h: 1.56 μg/mL 48h: 0.78 μg/mL *(C. albicans)*	Inhibit candidiasis infections	[113]
		N/A	Antifungal Antibiofilm	MIC 200 μg/mL *(C. albicans)*	Inhibit fungal infection	[7]
		Lactonic	Antimicrobial	MIC 97.5 mg/mL MBC ^9^ 195 mg/mL *(Streptococcus Oralis)*	Inhibit oral pathogens	[117]
		Lactonic	Antimicrobial	MIC 1 mg/mL *(L. fermentum)* 1.3 mg/mL *(L. acidophilus)*	Inhibit oral cariogenic bacteria	[118]
		Acidic and lactonic	Antimicrobial	MIC 19.5 μg/mL (Mixed culture of GPB and GNB)	Inhibit bacteria isolated from salted hides (Leather industry)	[119]
		Acidic and lactonic	Antimicrobial	MIC/MBC 2.09 μmol *(S.aureus)* 147 μmol *(E. coli)*	Inhibit both GPB and GNB	[123]
		Acidic and lactonic	Antimicrobial Antibiofilm	MIC_75_ ^10^ 0.8% *w/v* *(S. aureus)*	Inhibit biofilm formation of bacteria; applied for silicon catheter medical devices	[108]
		Acidic and lactonic	Antibiofilm	MIC 50 μg/mL *(S. aureus)*	Inhibit biofilm formation of bacteria; Applied for silicon catheter medical devices	[109]
		Acidic; lactonic (AS/LS ratio is 3.8:6.2)	Antibiofilm	CMC 0.1 wt% (*Pseudomonas* *aeruginosa* PAO1)	Inhibit biofilm formation of bacteria in microfluidic channels	[110]
		Acidic and lactonic	Antibiofilm	50 μg/mL *(P. aeruginosa)* 43.7% biofilm activity	Inhibit biofilm formation of bacteria; Applied in quorum quenching and imaging	[111]
-	*Cryptococcus sp.*	Acidic	Antimicrobial; Stabilizers for NPs production	5 mg/mL *(S. enterica* and * C. albicans)*	Inhibit growth of microbial cells, production of functionalized oxide nanoparticles (NPs)	[122]
	*Candida tropicalis* RA1	Lactonic	Antibacterial	MIC 250 μg/mL *(S. aureus)*	Inhibit GPB	[103]
	*Rhodotorula babjevae*	Lactonic	Antifungal	MIC 1 mg/mL *(Trichophyton mentagrophytes)*	Inhibit dermatophytosis	[115]
	N/A	Sophorolipid	Antifungal	0.1% *(Trichophyton mentagrophytes)*	Prevent tinea pedis	[114]
	N/A	Sophorolipid	Antiadhesion	MIC 100 μg/mL *(S.aureus, E. coli)*	Inhibit bacterial biofilm and adhesion to abiotic surfaces	[107]
Anticancer	*Starmerella* *bombicola*	C18:1 DLSL ^11^	Anticancer	IC_50_ 30 μg/mL (MDA-MB-231 breast cancer cells)	Inhibit breast cancer cells growth	[17]
			Anticancer	IC_50_ 12.23 μg/mL (HeLa cells) 25.45 μg/mL (CaSki cells)	Inhibit human cervical cancer cells growth	[125]
			Anticancer	IC_50_ ^12^ 70 μg/mL (HT-29 cells)	Inhibit colorectal cancer cells growth	[126]
		Lactonic	Anticancer	IC_50_ 14.14 μg/mL (HeLa cells)	Inhibit human cervical cancer cells growth	[127]
			Anticancer	IC_50_ 60 μg/mL (CT26 cells)	Inhibit colon cancer cells growth	[128]
		Acidic; lactonic (in excess)	Antiangiogenic	IC_50_ 63.89 µg/mL (EA.hy926 cells)	Inhibit angiogenesis of human cell lines (downregulating VEGF ^13^)	[130]
	*Wickerhamiella domercqiae*	C18:1 DLSL	Anticancer	IC_50_ 30 μg/mL (KYSE109 and KYSE450 cells)	Inhibit human esophageal cancer cells growth	[124]
	N/A	Lactonic	Anticancer	IC_50_ 25 μg/mL (HepG2 cells)	Induce apoptosis in liver hepatocellular carcinoma cells	[129]

^1^ C_22_-SL: sophorolipids containing 13-hydroxydocosanoic acid (OH-C_22_); ^2^ CAC: Critical Aggregation Concentration; ^3^ C_18”1_-NASL: Nonacetylated C_18”1_ acidic sophorolipids; ^4^ GPB: Gram-Positive Bacteria; ^5^ GNB: Gram-Negative Bacteria; ^6^ MIC_80_: Minimum Inhibitory Concentration required to inhibit the growth of 80% of organisms; ^7^ BIC_80_: Biofilm Inhibitory Concentration required to inhibit 80% metabolic activity of biofilm formation; ^8^ MIC_70_: Minimum Inhibitory Concentration required to inhibit the growth of 70% of organisms; ^9^ MBC: Minimum Biocidal Concentration; ^10^ MIC_75_: Minimum Inhibitory Concentration required to inhibit the growth of 75% of organisms; ^11^ C18:1 DLSL: Diacetylated lactonic sophorolipid with a C18 monounsaturated fatty acid; ^12^ IC_50_: Half maximal inhibitory concentration; ^13^ VEGF: Vascular endothelial growth factor. N/A: Not applicable.

## Data Availability

Not applicable.

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
