# Peer review of "Sophorolipids—Bio-Based Antimicrobial Formulating Agents for Applications in Food and Health"

_molecules, 2022, doi:10.3390/molecules27175556_

Round 1
Reviewer 1 Report
Dear all, Please, find attached Review results for the assigned manuscript. All comments are included in the manuscript. Overall, this is a good review paper that synthesizes various aspects of sophorolipids (SLs) and its applications. Most of the comments are grammatical / or editorial in nature. The abstract is O.K. and precise. The introduction is also adequate and written well. The other manuscript components such as overview of SLs (structurse & forms, bio-synthesis, biochemical pathways, regulatory mechanisms), methods for data compilation & synthesis, application of sophorolipids (food preservation, agricultural, bio-conversion of food waste, cosmetic & health) are also well presented and adequately addressed. The Tables are Figures are relevant and fit with the contents of the review paper. The sections of future perspectives and conclusions should be revised. The literature citations are also adequate. The various comments and points of clarification are directly inserted in the manuscript by using adobe tools for insertions / deletions & comments. Thank you.

Author Response
Dear Reviewer, please see the point-by-point response to the comments. The manuscript has been revised according to the suggestions.
|
Reviewer comments |
Responses |
|
L128, Is there any reason as to why they were associated with bumble bees? |
Starmerella bombicola was isolated from nectar of bumblebees, and S. bombicola is one of the sophorolipid producing yeast strain (L148) |
|
L549, Were the results better than the use of antibiotics alone? |
Yes, combined use of sophorolipid and antibiotics have lower MIC than using antibiotics alone. (L578) |
|
L750, This is a review paper, what kind of investigation you are talking about? |
Agree with the suggestion. This section has been revised (L761-764) |
Reviewer 2 Report
Sophorolipids are the most promising biosurfactants, have extensive applications in many fields. Many efforts have been focused on the biological activity of SLs recently. This study aims to examine current research trends of sophorolipids and evaluate their applications in food and health.
Some comments:
1. L40, fungi belong to yeast.
2. L57, the variations of SLs structures also include the unsaturation degree
3. L54, Fig. 2 only represent one structure of acidic SLs, while the lactonic type with variations on the acetylation degree, fatty acid chain length and unsaturated degree were not reflected.
4. L158-L164, the logic of this description needs to be improved.
5. L170, the difference between acidic SLs and lactonic SLs lies in whether the carboxyl group at the end of the fatty acid chain is esterified or not, rather than acetylated.
6. L217, acidic SLs is the bolaform SLs.
7. L332, the content of this paragraph is somewhat similar to the previous paragraph
8. L462, section 4.1.3, this section is not relevant to the application of SLs in food industry.
9. TABLE 1, SLBE is not the products by Starmerella bombicola but the SLs derivative
10. L691, some prospects related to the antimicrobial applications of SLs in food industry is encouraged.
Author Response
Dear Reviewer, please see the point-by-point response to the comments below. The manuscript has been revised according to the suggestions.
|
Reviewer comments |
Responses |
|
L40, fungi belong to yeast. |
Amended (L64) |
|
L57, the variations of SLs structures also include the unsaturation degree |
Amended (L82) |
|
L54, Fig. 2 only represent one structure of acidic SLs, while the lactonic type with variations on the acetylation degree, fatty acid chain length and unsaturated degree were not reflected. |
Amended; Figure 2 includes both acidic and lactonic group of SL (L174) |
|
L158-L164, the logic of this description needs to be improved. |
Amended; L180-186 |
|
L170, the difference between acidic SLs and lactonic SLs lies in whether the carboxyl group at the end of the fatty acid chain is esterified or not, rather than acetylated. |
Yes, this sentence is reflected in L190-193 |
|
L217, acidic SLs is the bolaform SLs. |
L240-243 has been rephrased to reflect the use of process development strategies in improving the yield of bolaform SL |
|
L332, the content of this paragraph is somewhat similar to the previous paragraph |
The paragraphs have been restructured (L345-346) |
|
L462, section 4.1.3, this section is not relevant to the application of SLs in food industry. |
Table 1 reflects the applications of sophorolipids in food industry and section 4.1.3 could represent part of the food waste management applications in the industry where sophorolipids can be extracted |
|
TABLE 1, SLBE is not the products by Starmerella bombicola but the SLs derivative |
Amended |
|
L691, some prospects related to the antimicrobial applications of SLs in food industry is encouraged. |
Yes, the antimicrobial applications of SL were included in L730-732 |
Reviewer 3 Report
-Figure 1 is loose in the text and is not referred to in the same.
-In lines 60 and 61 as well as in lines 73 and 74 repetitive phrase
-The authors cite reference 17 as recent, but it is from 2015 (7 years)
-The authors in this review seek to praise the use of sophorolipid based on published works. It would be interesting to offer the reader how much is currently produced, since the authors propose the use of this metabolite.
-Proponents need to beware of repeated information in the text
Author Response
Dear reviewer, please see the point-by-point response to the comments below. The manuscript has been revised according to the suggestions.
|
Reviewer comments |
Responses |
|
Figure 1 is loose in the text and is not referred to in the same. |
Amended; Figure 1 caption have been rephrased |
|
In lines 60 and 61 as well as in lines 73 and 74 repetitive phrase |
Amended; repetitive phrases have been removed |
|
The authors cite reference 17 as recent, but it is from 2015 (7 years) |
Amended; the term recent have been removed |
|
The authors in this review seek to praise the use of sophorolipid based on published works. It would be interesting to offer the reader how much is currently produced, since the authors propose the use of this metabolite |
We have seen significant increase in the interest in production of sophorolipids in published work (based on research databases) over the past 5 years but the actual production at the industrial scale in respective locations internationally may not be reflected at this point in our review. |
|
Proponents need to beware of repeated information in the text |
Yes the manuscript has been revised to avoid repeated content.
|
Round 2
Reviewer 3 Report
Figure 2 was you who made or took from some reference. If it was from any reference, report it in the Legend
Figure 3 the same
Author Response
Dear reviewer, thank you for your comments. Both Figure 2 and 3 were prepared by the authors.